# Development and validation of artificial intelligence to detect and diagnose liver lesions from ultrasound images

Thodsawit Tiyarattanachai[1], Terapap Apiparakoon[2], Sanparith Marukatat[3], Sasima Sukcharoen[4], Nopavut Geratikornsupuk[5], Nopporn Anukulkarnkusol[6], Parit Mekaroonkamol[4], Natthaporn Tanpowpong[7], Pamornmas Sarakul[8], Rungsun Rerknimitr[9], Roongruedee Chaiteerakij[9]*

1 Faculty of Medicine, Chulalongkorn University, Bangkok, Thailand, 2 Department of Computer Engineering, Faculty of Engineering, Chulalongkorn University, Bangkok, Thailand, 3 Image Processing and Understanding Team, Artificial Intelligence Research Group, National Electronics and Computer Technology Center, Pathum Thani, Thailand, 4 Division of Gastroenterology, Department of Medicine, King Chulalongkorn Memorial Hospital, The Thai Red Cross Society, Bangkok, Thailand, 5 Department of Medicine, Queen Savang Vadhana Memorial Hospital, The Thai Red Cross Society, Chonburi, Thailand, 6 Gastroenterology and Liver Diseases Center, Mahachai Hospital, Samut Sakhon, Thailand, 7 Department of Radiology, Faculty of Medicine, Chulalongkorn University and King Chulalongkorn Memorial Hospital, Bangkok, Thailand, 8 Department of Radiology, Mahachai Hospital, Samut Sakhon, Thailand, 9 Division of Gastroenterology, Department of Medicine, Center of Excellence for Innovation and Endoscopy in Gastrointestinal Oncology, Faculty of Medicine, Chulalongkorn University, Bangkok, Thailand

* roon.chaiteerakij@chula.md

**Data Availability Statement:** All relevant data are within the manuscript and its Supporting Information files.

## Abstract

Artificial intelligence (AI) using a convolutional neural network (CNN) has demonstrated promising performance in radiological analysis. We aimed to develop and validate a CNN for the detection and diagnosis of focal liver lesions (FLLs) from ultrasonography (USG) still images. The CNN was developed with a supervised training method using 40,397 retrospectively collected images from 3,487 patients, including 20,432 FLLs (hepatocellular carcinomas (HCCs), cysts, hemangiomas, focal fatty sparing, and focal fatty infiltration). AI performance was evaluated using an internal test set of 6,191 images with 845 FLLs, then externally validated using 18,922 images with 1,195 FLLs from two additional hospitals. The internal evaluation yielded an overall detection rate, diagnostic sensitivity and specificity of 87.0% (95%CI: 84.3–89.6), 83.9% (95%CI: 80.3–87.4), and 97.1% (95%CI: 96.5–97.7), respectively. The CNN also performed consistently well on external validation cohorts, with a detection rate, diagnostic sensitivity and specificity of 75.0% (95%CI: 71.7–78.3), 84.9% (95%CI: 81.6–88.2), and 97.1% (95%CI: 96.5–97.6), respectively. For diagnosis of HCC, the CNN yielded sensitivity, specificity, and negative predictive value (NPV) of 73.6% (95% CI: 64.3–82.8), 97.8% (95%CI: 96.7–98.9), and 96.5% (95%CI: 95.0–97.9) on the internal test set; and 81.5% (95%CI: 74.2–88.8), 94.4% (95%CI: 92.8–96.0), and 97.4% (95%CI: 96.2–98.5) on the external validation set, respectively. CNN detected and diagnosed common FLLs in USG images with excellent specificity and NPV for HCC. Further development of an AI system for real-time detection and characterization of FLLs in USG is warranted.

**Funding:** This research project is supported by The Second Century Fund (C2F), Chulalongkorn University (TT); Ratchadapisek Sompoch Endowment Fund (2019) under Telehealth Cluster, Chulalongkorn University (RC); and Grant for International Research Integration: Chula Research Scholar, Ratchadaphiseksomphot Endowment Fund, Chulalongkorn University (RR). The funders had no role in study design, data collection and analysis, decision to publish, or preparation of the manuscript.

**Competing interests:** The authors have declared that no competing interests exist.

**Abbreviations:** 95% CI, 95% confidence interval; AI, artificial intelligence; CNN, convolutional neural network; FFI, focal fatty infiltration; FFS, focal fatty sparing; FLL, focal liver lesion; FN, false negative; FP, false positive; IoU, Intersection-over-Union; IQR, interquartile range; KCMH, King Chulalongkorn Memorial Hospital; NPV, negative predictive value; PACS, Picture Archiving and Communication System; PPV, positive predictive value; TN, true negative; TP, true positive; USG, ultrasonography.

## Introduction

Hepatocellular carcinoma (HCC) is the fourth leading cause of cancer death worldwide [1]. Screening abdominal ultrasonography (USG) has been shown to be cost-effective in reducing mortality from hepatocellular carcinoma (HCC) by 37% [2–5]. However, worldwide surveillance rates remain low, ranging from 6.7–28.0% [5–9]. One significant barrier to timely HCC screening is inaccessibility to high-quality ultrasound with interpreting radiologists, particularly in rural areas [10]. Developing an artificial intelligence (AI)-assisted USG image analysis system may potentially facilitate USG screening programs, increase the surveillance rate and improve the survival of HCC patients.

AI systems have shown potential in facilitating radiologic image interpretation [11]. Abdominal USG is one of the most challenging imaging modalities in the field of AI-based medical image analysis for several reasons. First, the quality of USG images varies among different devices and operators [12]. Second, USG images have a low signal-to-noise ratio making the identification of small lesions from the background difficult. Additionally, a single abdominal USG image often contains several organ structures, often including the liver, gallbladder, kidney, bile duct and blood vessels. The position and orientation of these structures in USG images are not consistent and standardized as with CT or MRI images, therefore, differentiating true lesions from normal structures and pseudo-lesions can be challenging. Although previous studies on AI neural networks demonstrated 88–96% accuracy in the diagnosis of focal liver lesions (FLLs) in still USG images, the size of the training datasets were small with only internal tests being performed [13–15]. Whether these AI systems can be applied in other clinical settings has yet to be investigated.

In the current study, we used a large number of off-line USG images to develop an AI-assisted USG image analysis system for detection and diagnosis of various FLLs including HCC, cyst, hemangioma, focal fatty sparing (FFS), and focal fatty infiltration (FFI). To strengthen generalizability of our AI system, we evaluated its performance on images from both an internal test set and external validation datasets (i.e. images from different hospitals using different machines and different sonographers).

## Materials and methods

### Dataset

This retrospective study was approved by the Institutional Review Board of the Faculty of Medicine, Chulalongkorn University (IRB No. 423/61 and 646/62). Data was collected upon approval from the director and/or ethics committee of King Chulalongkorn Memorial Hospital, Bangkok, Thailand; Mahachai Hospital, Samut Sakhon, Thailand; and Queen Savang Vadhana Memorial Hospital, Chonburi, Thailand. Requirement for informed consent was waived due to the retrospective nature of this study. All ultrasound examinations were de-identified and analyzed anonymously. Images from upper abdominal USG performed between 2010 and 2019 were retrospectively retrieved from the Picture Archiving and Communication System (PACS) of three different hospitals. All data were still images taken as snapshots during ultrasound. They had been stored in Digital Imaging and Communications in Medicine (DICOM) format. All images were acquired using curvilinear transducers and allocated into 3 datasets: training set, internal test set and external validation set. The training set and the internal test set were retrieved from the same patient batch at the main study site, King Chulalongkorn Memorial Hospital, Bangkok, Thailand. All images from this batch were randomly allocated in a 9:1 ratio of the training set to the internal test set. Allocation design ensured that all images from the same patient were assigned to the same set making the image sets independent of

each other without any duplicated patients. The external validation set was acquired from Mahachai Hospital, Samut Sakhon, Thailand and Queen Savang Vadhana Memorial Hospital, Chonburi, Thailand to further validate the performance of the AI system. The external validation images were completed by different sonographers using a variety of USG machine models. We included USG studies with all ranges of image qualities from new and older machines to ensure that the AI system can be generalized to other datasets. A total of 17 different ultrasound machine models were included in this study (**S1 Table in S1 File**).

Five of the most commonly encountered liver lesions, including HCCs, cysts, hemangiomas, FFSs and FFIs were selected for this study (**Fig 1**) [16, 17]. The definitive diagnoses of FLLs were verified using pathology and/or MRI/CT reports. Pathology reports were reviewed first. If not available, MRI and CT reports were then considered. Exclusion criteria were USG studies without further investigation for definitive diagnoses of FLLs and USG studies in which the lesion characteristics were altered by prior treatments. It is noted that in each USG study, there were images with and without FLLs. The normal images without FLLs, which were randomly selected in a 1:1 ratio, were used as negative controls for training the AI system to learn to distinguish FLLs from normal organ structures. An equal number of both types of images facilitated the training process for AI to correctly detect FLLs while minimizing false positivity. In contrast, for the internal test set and external validation set, all negative control images were included in order to replicate the real-world situation in which rare instances of FLLs emerge among a vast number of images showing normal liver and other normal organs.

Since some patients had more than 1 USG study and some studies had more than one image containing FLLs, the following protocol was used to select and include images in the dataset. For the training set, we included all FLL images of all USG studies of each patient in order to diversify images for the AI training. By contrast, in the internal test set and external validation set, we included up to 2 images with FLLs per study and up to 2 studies per patient. For the USG study having >1 image with FLLs, 2 images containing different FLLs were randomly selected. If there were >1 image containing the same FLL, 2 images taken at different probe positions were randomly selected. If there were >1 image with the same FLL taken at identical probe position, only 1 image was randomly selected.

## AI system architecture

The AI framework used in this study was a convolutional neural network (CNN) [18]. CNNs are currently the preferred technique for several types of image analyses due to its structured layering characteristic that can detect complex features of the input images, where the shallow layers detect simple features such as dots and lines and the deeper layers detect more complex features, such as curves and loops [18]. In the present study, we adopted a CNN architecture called RetinaNet [19] which takes an image as input and creates a set of bounding boxes surrounding the FLL along with its class (predicted diagnosis) and its confidence in predicting that particular diagnosis. Confidence value range from 0 to 1, with a value of 1 being the most confident. The confidence threshold can be adjusted according to clinical relevance; for example, the confidence threshold may be lowered to increase the detection rate for HCC if needed in a certain patient population. The overall performance of the CNN, therefore, varies by different confidence thresholds. In this study, the confidence threshold was selected such that the F2 score was optimized on a tuning set, which was a subset of the training set (Details in **S3 Appendix and S8 Fig in S1 File**). The selected confidence threshold was then used in both the internal test set and external validation cohorts. Since each diagnosis was independent, it was possible for RetinaNet to output multiple diagnoses for a single lesion. This approach resembled the usual practice of reporting differential diagnoses of FLLs by radiologists.

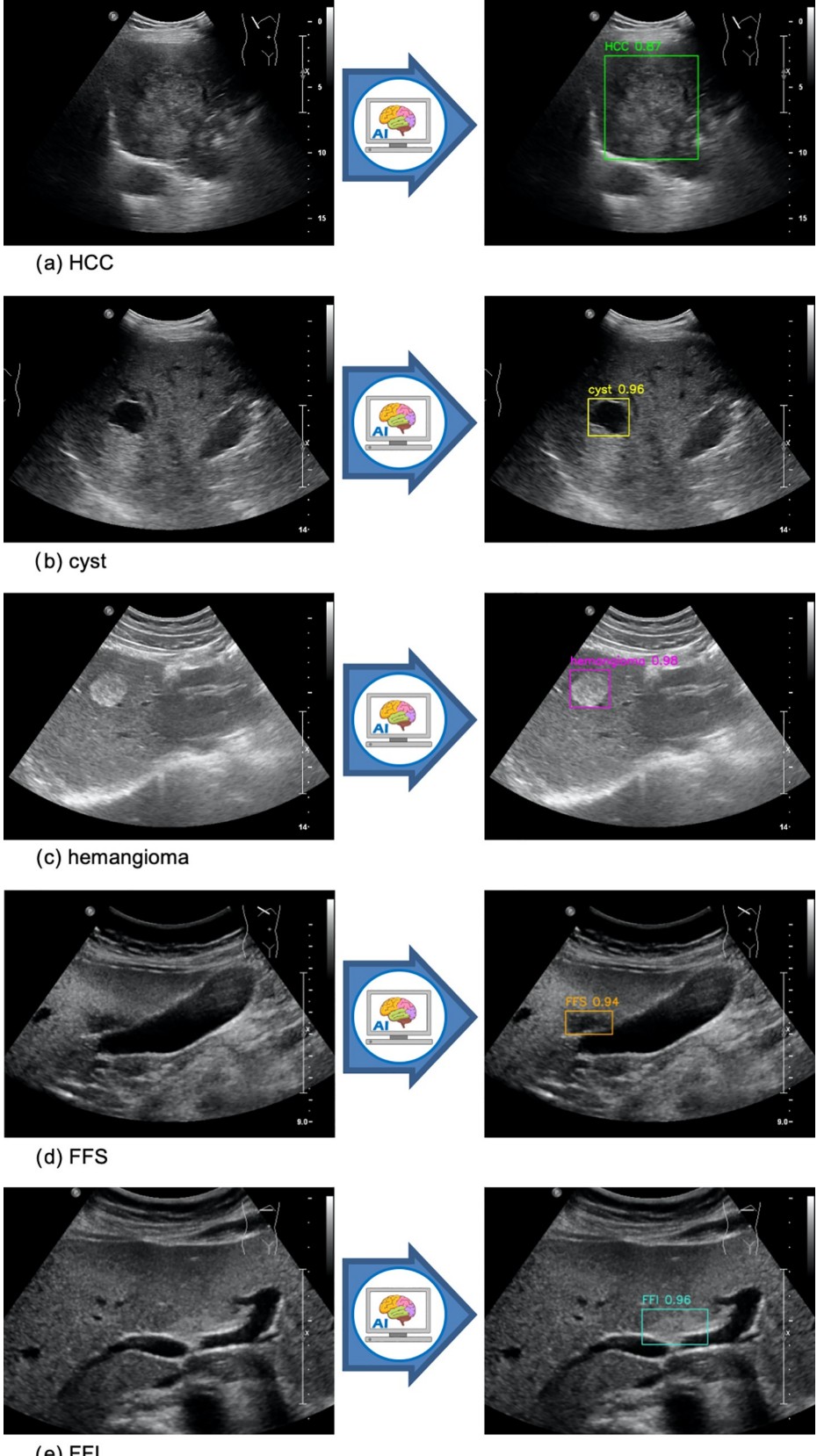

(a) HCC

(b) cyst

(c) hemangioma

(d) FFS

(e) FFI

**Fig 1.** Example images of HCC (1a), cyst (1b), hemangioma (1c), FFS (1d), and FFI (1e) included in the study. Left panels show original images inputted into the AI system. Right panels show AI-outputted bounding boxes around each lesion along with the predicted diagnoses and its confidence value for prediction.

## Ultrasound image preprocessing

During image preprocessing, all patient identification information and the peripheral areas in the USG images were cropped out. We identified the coordinates of fan-shaped USG region by 'Sequence of Ultrasound Regions' DICOM header, in order to ensure that the cropped image contained only the fan-shaped USG region where annotations and dimension measurements had been cropped out. We also removed markers which were made by sonographers in some images (**S1 Appendix in S1 File**). The images were then resized to 1333 pixels wide and 800 pixels tall before inputted into the CNN.

## AI system development process

**Training phase.** A supervised training method was implemented to train the AI system. In order to generate an image training dataset, pathology and/or MRI/CT reports were reviewed by experienced sonographers to identify labels, which were the locations and definitive diagnoses of FLLs in each USG image [20]. A hepatologist (R.C.) subsequently verified the labels to ensure their accuracy. Images in the training set were fed into the AI system to train it to predict the location and diagnosis of the FLLs (**Fig 2**).

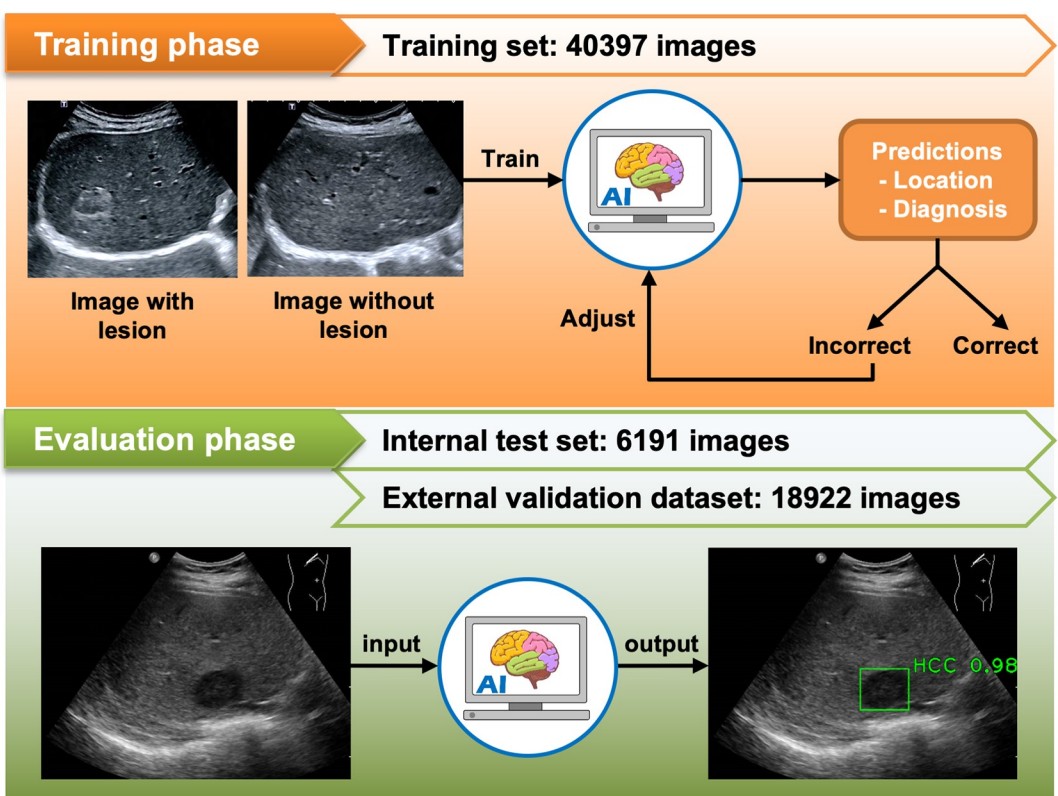

**Fig 2. Process of AI system training and evaluation.**

RetinaNet codes were adopted from an open-source repository [21, 22]. The codes were then modified and optimized for analyzing USG images. In this work, RetinaNet was composed of backbone ResNet50 and the detection and diagnosis heads. The backbone ResNet50 extracted the hierarchy of features, and the detection and diagnosis heads subsequently processed these features and outputted locations and diagnoses of FLLs [23].

The training was done in two main steps. First, the backbone ResNet50 was trained on a publicly-available image dataset called Microsoft Common Objects in Context (MS-COCO), which comprises 330,000 images of 1.5 million object instances [24]. Subsequently, the whole CNN, both backbone and heads, was fine-tuned on our USG images in the training set. The CNN was trained for 500,000 iterations (25 epochs × 20,000 steps per epoch) on USG images. The initial learning rate was 0.0001. To enable the CNN to recognize diverse configurations of images and to maximize the number of training images, image augmentation was performed by horizontal translation, vertical translation, rotation, scaling, horizontal flipping, motion blur, contrast, brightness, hue and saturation adjustment at each iteration [25]. The training hyperparameters are shown in the **S8 Table in S1 File**. During training, a tuning set was used to monitor performance of the CNN. We selected an epoch that optimized mean average precision [26] on the tuning set for final evaluation on the internal test set and the external validation set.

**Evaluation phase.** The performance of the developed AI system was evaluated first on the internal test set, then on the external validation set.

## Performance evaluation metrics

We separately evaluated detection and diagnosis, the two primary tasks of the CNN. Evaluation of detection rates and diagnosis performance were performed on a per-lesion basis. The definitions of the evaluation metrics are described below.

**Detection task.** An FLL was counted as correctly detected if the CNN generated a bounding box around it and the box overlapped with the true location of FLL, which was assessed using Intersection-over-Union (IoU). In this study, an IoU of greater than 0.2 was a cut-off for a correct detection by the CNN (**S2 Appendix and S1 Fig in S1 File**). We opted to use this cut-off because FLLs in USG images often have indistinct boundaries, especially for FFSs and FFIs (**S2 Fig in S1 File**). The detection rate was calculated by dividing the number of FLLs correctly detected by the number of total FLLs. Detection rates stratified by ground truth diagnoses were also evaluated. In contrast, a false positive detection was counted when the AI system outputted a bounding box on an area that did not contain FLLs (e.g. liver parenchyma, normal organ structures, etc.). Evaluation of false positive detections was performed on a per-image basis.

**Diagnosis task.** We used the following metrics to evaluate AI diagnostic performance:

$$sensitivity = \frac{TP}{TP + FN}$$

$$specificity = \frac{TN}{TN + FP}$$

$$accuracy = \frac{TP + TN}{TP + TN + FP + FN}$$

$$positive\ predictive\ value = \frac{TP}{TP + FP}$$

$$negative\ predictive\ value = \frac{TN}{TN + FN}$$

where TP, TN, FP and FN are the number of true positive, true negative, false positive and false negative diagnoses, respectively.

We used a "one-versus-all" method to evaluate diagnostic performance for each FLL diagnosis [27]. For example, when evaluating diagnostic performance for HCC, other diagnoses were counted as a single non-HCC class:

$$sensitivity_{HCC} = \frac{TP_{HCC}}{TP_{HCC} + FN_{HCC}}$$

where $sensitivity_{HCC}$ is the diagnostic sensitivity for HCC.

$TP_{HCC}$ is the number of true positive diagnoses for HCC, where the definitive diagnosis is HCC and the AI system correctly diagnosed the lesion as HCC.

$FN_{HCC}$ is the number of false negative diagnoses for HCC, where the definitive diagnosis is HCC, but the AI system falsely diagnosed the lesion as either cyst, hemangioma, FFS or FFI.

In cases where multiple diagnoses reached the confidence threshold and hence were predicted by the AI system, only the diagnosis with the highest confidence value was selected as the AI prediction.

**Calculation of overall detection rate and overall diagnostic performance.** After calculating detection and diagnostic performance metrics for each definitive diagnosis of FLLs, we pooled the performance results from the 5 FLL diagnoses into an overall performance result. Because the numbers of each FLL diagnosis in our dataset were imbalanced, overall performance, including overall detection rate, overall diagnostic sensitivity and specificity, were pooled by an unweighted average, to minimize the effect of imbalanced number of FLL diagnoses. For example,

$$DR_{overall} = \frac{DR_{HCC} + DR_{cyst} + DR_{hemangioma} + DR_{FFS} + DR_{FFI}}{5}$$

where $DR_{overall}$ is the overall detection rate.

$DR_{HCC}$, $DR_{cyst}$, $DR_{hemangioma}$, $DR_{FFS}$ and $DR_{FFI}$ are the detection rates for HCC, cyst, hemangioma, FFS and FFI, respectively.

$$sens_{overall} = \frac{sens_{HCC} + sens_{cyst} + sens_{hemangioma} + sens_{FFS} + sens_{FFI}}{5}$$

where $sens_{overall}$ is the overall diagnostic sensitivity.

$sens_{HCC}$, $sens_{cyst}$, $sens_{hemangioma}$, $sens_{FFS}$ and $sens_{FFI}$ are the diagnostic sensitivities for HCC, cyst, hemangioma, FFS and FFI, respectively.

## Statistical analysis

Performance of the CNN was reported by detection rates, false positive detection rates, diagnostic sensitivities, specificities, accuracies, positive predictive values (PPVs), and negative predictive values (NPVs) with 95% confidence intervals (95% CI). Detection and diagnostic performance of each type of FLL as well as overall performance for all FLL diagnoses were calculated. Performance on the internal test set and external validation set were compared using two-tailed z-test for difference of proportion. Python version 3.7 (Python Software Foundation, Delaware, USA) and IBM SPSS Statistics for Windows, version 22 (SPSS Inc., Chicago,

**Table 1. Number of USG images from 3 participating hospitals, along with allocation of images for AI training and performance evaluation.**

| | Training set[a] | Internal test set[a] | External validation cohorts | | |
| --- | --- | --- | --- | --- | --- |
| | | | Cohort 1[b] | Cohort 2[c] | Pooled |
| **Number of patients** | 3487 | 385 | 311 | 625 | 936 |
| **Total images** | 40397 | 6191 | 5624 | 13298 | 18922 |
| **Total images with FLLs** | 18239 | 801 | 344 | 734 | 1078 |
| **Number of lesions (%)** | | | | | |
| Total | 20432 (100) | 845 (100) | 360 (100) | 835 (100) | 1195 (100) |
| HCC | 2414 (11.8) | 102 (12.1) | 34 (9.4) | 104 (12.5) | 138 (11.5) |
| Cyst | 6600 (32.3) | 215 (25.4) | 130 (36.1) | 87 (10.4) | 217 (18.2) |
| Hemangioma | 5374 (26.3) | 217 (25.7) | 60 (16.7) | 202 (24.2) | 262 (21.9) |
| FFS | 5110 (25.0) | 264 (31.2) | 120 (33.3) | 404 (48.4) | 524 (43.8) |
| FFI | 934 (4.6) | 47 (5.6) | 16 (4.4) | 38 (4.6) | 54 (4.5) |
| **Median sizes in cm (IQR)** | | | | | |
| Total | 1.6 (1.7) | 1.6 (1.6) | 1.5 (1.3) | 1.8 (1.7) | 1.7 (1.6) |
| HCC | 3.7 (5.5) | 3.3 (5.8) | 2.3 (6.6) | 3.9 (4.4) | 3.9 (5.5) |
| Cyst | 1.4 (1.5) | 1.0 (0.8) | 1.0 (0.7) | 1.2 (0.9) | 1.1 (0.9) |
| Hemangioma | 1.2 (1.2) | 1.4 (1.1) | 1.9 (1.1) | 1.5 (1.5) | 1.6 (1.4) |
| FFS | 1.7 (1.1) | 1.8 (1.4) | 1.9 (1.4) | 1.7 (1.3) | 1.8 (1.3) |
| FFI | 2.5 (2.5) | 2.4 (3.5) | 1.7 (1.0) | 2.4 (2.7) | 2.1 (2.7) |
| **Total images without FLLs** | 22158 | 5390 | 5280 | 12564 | 17844 |

[a]King Chulalongkorn Memorial Hospital, Bangkok, Thailand

[b]Mahachai Hospital, Samut Sakhon, Thailand

[c]Queen Savang Vadhana Memorial Hospital, Chonburi, Thailand

Ill., USA) were used for data analyses. A p-value of <0.05 was considered statistically significant.

## Results

### Baseline characteristics

A total of 40,397 images with 20,432 FLLs were included in the training set, while 6,191 images with 845 FLLs and 18,922 images with 1,195 FLLs were included in the internal test set and external validation set, respectively. Baseline characteristics of each dataset is described in **Table 1**.

### Performance of the CNN

Performance of CNN in detection and diagnosis on the internal test set and external validation set are summarized in **Table 2**.

**Lesion detection performance.** On the internal test set, the CNN had an overall lesion detection rate of 87.0% (95%CI: 84.3–89.6). The median IoU was 0.788 (range: 0.202–0.978) (**S3 Fig in S1 File**), suggesting an exceptional agreement between the predicted and true location of the FLL. Compared to the internal test set, the overall detection rate in the pooled external validation set was significantly lower (75.0% (95%CI: 71.7–78.3), p < 0.001), with the median IoU of 0.781 (range: 0.201–0.970) (**S3 Fig in S1 File**).

The false positive detection rate was 3.7% (226/6191) and 5.1% (970/18922) in the internal test set and external validation set, respectively. The images with false positive detections were reviewed. Blood vessel in the liver was the most common falsely identified structure as FLLs

**Table 2. Performance of the AI system on the internal test set and external validation cohorts.**

| | Internal test set[a] | External validation cohorts | | | P[*] |
|---|---|---|---|---|---|
| | | Cohort 1[b] | Cohort 2[c] | Pooled | |
| **Overall** | | | | | |
| Detection rate | 87.0 (84.3–89.6) | 80.3 (74.8–85.8) | 73.9 (69.9–78.0) | 75.0 (71.7–78.3) | <0.001 |
| Diagnostic sensitivity | 83.9 (80.3–87.4) | 84.6 (79.0–90.2) | 85.7 (81.7–89.6) | 84.9 (81.6–88.2) | 0.69 |
| Diagnostic specificity | 97.1 (96.5–97.7) | 97.2 (96.3–98.2) | 97.0 (96.3–97.7) | 97.1 (96.5–97.6) | 0.98 |
| **HCC** | | | | | |
| Detection rate | 85.3 (78.4–92.2) | 91.2 (81.6–101) | 74.0 (65.6–82.5) | 78.3 (71.4–85.2) | 0.16 |
| Diagnostic sensitivity | 73.6 (64.3–82.8) | 74.2 (58.8–89.6) | 84.4 (76.3–92.5) | 81.5 (74.2–88.8) | 0.19 |
| Diagnostic specificity | 97.8 (96.7–98.9) | 96.1 (93.7–98.5) | 93.6 (91.5–95.7) | 94.4 (92.8–96.0) | 0.55 |
| **Cyst** | | | | | |
| Detection rate | 89.3 (85.2–93.4) | 76.9 (69.7–84.2) | 85.1 (77.6–92.5) | 80.2 (74.9–85.5) | 0.008 |
| Diagnostic sensitivity | 97.9 (95.9–99.9) | 91.0 (85.4–96.6) | 98.6 (96.0–100) | 94.3 (90.8–97.7) | 0.07 |
| Diagnostic specificity | 98.3 (97.2–99.4) | 97.8 (95.8–99.9) | 98.7 (97.7–99.7) | 98.5 (97.6–99.4) | 0.99 |
| **Hemangioma** | | | | | |
| Detection rate | 93.5 (90.3–96.8) | 78.3 (67.9–88.8) | 79.7 (74.2–85.2) | 79.4 (74.5–84.3) | <0.001 |
| Diagnostic sensitivity | 80.8 (75.4–86.2) | 74.5 (62.0–86.9) | 67.7 (60.5–74.9) | 69.2 (63.0–75.5) | 0.006 |
| Diagnostic specificity | 95.0 (93.2–96.9) | 97.9 (96.1–99.7) | 96.2 (94.4–98.0) | 96.8 (95.5–98.1) | 0.12 |
| **FFS** | | | | | |
| Detection rate | 77.3 (72.2–82.3) | 80.0 (72.8–87.2) | 67.6 (63.0–72.1) | 70.4 (66.5–74.3) | 0.03 |
| Diagnostic sensitivity | 98.0 (96.1–99.9) | 100 (96.2–100)[†] | 98.5 (97.1–100) | 98.9 (97.9–100) | 0.41 |
| Diagnostic specificity | 96.9 (95.5–98.4) | 95.8 (92.9–98.6) | 98.5 (97.2–99.8) | 97.5 (96.2–98.9) | 0.53 |
| **FFI** | | | | | |
| Detection rate | 89.4 (80.5–98.2) | 75.0 (53.8–96.2) | 63.2 (47.8–78.5) | 66.7 (54.1–79.3) | 0.004 |
| Diagnostic sensitivity | 69.0 (55.1–83.0) | 83.3 (62.2–100) | 79.2 (62.9–95.4) | 80.6 (67.6–93.5) | 0.60 |
| Diagnostic specificity | 97.4 (96.2–98.6) | 98.5 (97.1–100) | 98.1 (97.0–99.2) | 98.3 (97.4–99.1) | 0.97 |

[a]KCMH, King Chulalongkorn Memorial Hospital, Bangkok, Thailand

[b]Mahachai Hospital, Samut Sakhon, Thailand

[c]Queen Savang Vadhana Memorial Hospital, Chonburi, Thailand

[*]P-value for two-tailed z-test for difference of proportion, comparing performance results in the internal test set and pooled external validation set. P-value of <0.05 was considered statistically significant.

[†]Clopper-Pearson confidence interval was calculated for performance value at boundaries (i.e. 0% and 100%)

Detection rates, diagnostic sensitivities and specificities are shown in percentages. 95% confidence intervals are shown in parenthesis.

(12.3%, 147/1196), followed by heterogeneous background liver parenchyma (7.4%, 88/1196), renal cysts (6.8%, 81/1196), inferior vena cava (3.4%, 41/1196) and splenic lesions (2.8%, 33/1196) (**S2 Table and S4 Fig in S1 File**). Likewise, 114 and 273 images with false negative detection in the internal test set and the external validation set were reviewed. Characteristics for incorrect detection included being a small lesion <1 cm (27.4%), having an uncommon location of that particular diagnosis (8.0%), lesion with atypical characteristics (7.8%), ill-defined lesion (7.5%), and lesion obscured by shadow artifacts or not completely visible (6.2%) (**S3 Table and S5 Fig in S1 File**).

**Diagnostic performance.** After detection of a lesion, the AI algorithm identified and diagnosed the lesion as one of five diagnoses (HCC, cyst, hemangioma, FFS, FFI) of FLLs. On the internal test set, the CNN had overall sensitivity, specificity, accuracy, PPV and NPV of 83.9% (95%CI: 80.3–87.4), 97.1% (95%CI: 96.5–97.7), 95.4% (95%CI: 94.8–96.1), 83.6% (95% CI: 80.1–87.1), and 97.2% (95%CI: 96.6–97.8), respectively, for classifying any FLLs. For the

**Table 3. Confusion matrix for classification results on internal test set and external validation set.**

| Internal test set | | | | | | | |
|---|---|---|---|---|---|---|---|
| | | Definitive diagnosis | | | | | |
| | | HCC | Cyst | Hemangioma | FFS | FFI | Total |
| **Predicted diagnosis by AI** | HCC | 64 | 2 | 9 | 1 | 2 | 78 |
| | Cyst | 3 | 188 | 4 | 2 | 0 | 197 |
| | Hemangioma | 13 | 2 | 164 | 1 | 10 | 190 |
| | FFS | 5 | 0 | 10 | 200 | 1 | 216 |
| | FFI | 2 | 0 | 16 | 0 | 29 | 47 |
| | Total | 87 | 192 | 203 | 204 | 42 | 728 |

| Pooled external validation set | | | | | | | |
|---|---|---|---|---|---|---|---|
| | | Definitive diagnosis | | | | | |
| | | HCC | Cyst | Hemangioma | FFS | FFI | Total |
| **Predicted diagnosis by AI** | HCC | 88 | 3 | 38 | 2 | 1 | 132 |
| | Cyst | 4 | 164 | 7 | 0 | 0 | 175 |
| | Hemangioma | 12 | 2 | 144 | 2 | 6 | 166 |
| | FFS | 3 | 5 | 5 | 365 | 0 | 378 |
| | FFI | 1 | 0 | 14 | 0 | 29 | 44 |
| | Total | 108 | 174 | 208 | 369 | 36 | 895 |

diagnosis of HCC, CNN had a sensitivity of 73.6% (95%CI: 64.3–82.8), specificity of 97.8% (95%CI: 96.7–98.9), accuracy of 94.9% (95%CI: 93.3–96.5), PPV of 82.1% (95%CI: 73.5–90.6), and NPV of 96.5% (95%CI: 95.0–97.9). The sensitivity and specificity for diagnosing other FLLs ranged from 69.0% to 98.0% and 95.0% to 98.3%, respectively (**Table 2**).

The overall performance of the CNN in diagnosing any FLLs in the external validation set was similar to that of the internal test set, with the sensitivity, specificity, accuracy, PPV and NPV of 84.9% (95%CI: 81.6–88.2), 97.1% (95%CI: 96.5–97.6), 95.3% (95%CI: 94.7–95.9), 81.9% (95%CI: 78.4–85.4), and 97.1% (95%CI: 96.6–97.7), respectively. In subgroup analyses of each type of FLL, the diagnostic performance in the external validation set was also comparable to the performance of the internal test set as displayed in **Table 2**.

Confusion matrix for classification results in the internal test set and external validation set is shown in **Table 3**. After reviewing misclassified images, we found that the most common cause was atypical characteristics of FLLs (30.1%, 56/186) (**S4 Table and S6 Fig in S1 File**).

**Subgroup analyses.** The AI system detection and diagnostic performance was further stratified by FLL sizes (**S5 Table in S1 File**) and background liver parenchyma (cirrhosis vs. non-cirrhosis) (**S6 Table in S1 File**). As expected, diagnostic sensitivities for HCC increased by size. Sensitivities of HCC sizes of < 2 cm, 2–3 cm, and > 3 cm were 23.5% (95%CI: 3.4–43.7), 77.3% (95%CI: 59.8–94.8) and 89.6% (95%CI: 80.9–98.2) in the internal test set and 50.0% (95%CI: 30.0–70.0), 84.2% (95%CI: 67.8–100) and 92.3% (95%CI: 85.8–98.8) in the external validation set, respectively. Additionally, detection rates of HCC in cirrhosis subgroup were lower than in non-cirrhosis subgroup, i.e. 79.5% (95%CI: 67.6–91.5) vs 89.7% (95%CI: 81.8–97.5) in the internal test set and 72.0% (95%CI: 63.2–80.8) vs 94.7% (95%CI: 87.6–100) in the external validation set, respectively.

## Discussion

The CNN developed in our study using an advanced structured AI learning system demonstrated a consistently high diagnostic performance on USG images from both an internal test

set and an external validation set. It achieved overall diagnostic sensitivity and specificity of 83.9% and 97.1% on the internal test set and 84.9% and 97.1% on the external validation set.

Regarding detection task, our AI system was able to detect 85.3% of HCCs in the internal test set and 78.3% in the external validation set (p = 0.16). However, averaging all included FLL diagnoses, the detection rate of the external validation set was significantly lower than the internal test set (75.0% vs 87.0%; p <0.001). Factors that may be at play include the increased heterogeneity of image characteristics from different ultrasound machine models in the external validation set, compared to the training set (**S1 Table in S1 File**). This finding underscores the importance of image diversity in the training dataset. To enhance practicality, we propose to train the AI system with additional USG videos which contain numerous image frames to better detect FLLs.

For the diagnosis task, the performance results were consistent between the internal test set and the external validation set. The AI system achieved overall sensitivities of 83.9% and 84.9%, and specificities of 97.1% and 97.1% on the internal test set and external validation set, respectively. Our AI system had lower sensitivity for FLL diagnosis than the sensitivities of 93.8%-98.8% shown in previous studies, with comparable specificities of 94.3–98.9% in the previous reports [13–15]. The lower sensitivity may have been due to the wider spectrum of FLL diagnoses and characteristics. In the two previous studies, only HCCs, cysts and hemangiomas were selected for testing [14, 15]. In the current study, FFSs and FFIs were additionally included as both diagnoses are encountered frequently in liver cancer surveillance settings with prevalence rates of FFS and FFI previously reported as 6.3% and 9.2%, respectively [17, 28].

Misclassifications of FLLs by the AI system may be explained by the fact that different types of FLLs can appear very similar on USG images. Moreover, some lesions may have atypical characteristics. We found that HCCs and hemangiomas were sometimes interchangeably misdiagnosed (**Table 3**). This may be because our sample contained a considerable number of hemangiomas with atypical characteristics (18.8% of all hemangiomas) with 11.8% of hemangiomas appearing as hypoechoic lesions in fatty liver background and 7% of hemangiomas as giant hemangioma with heterogeneous echogenicity in contrast to typical well-defined round hyperechoic lesion (**S6 Fig in S1 File**). This is supported by our findings that diagnostic sensitivity of HCC increased when the size of lesion increased, while diagnostic sensitivities of hemangioma decreased when the size of lesion increased. We specifically had designed our AI system to output diagnoses of detected FLLs as differential diagnoses. This should be clinically useful as physicians will be able to decide what is the most likely diagnosis of FLL by incorporating the AI diagnosis together with their clinical information. We further analyzed whether HCC appeared in the top-k predicted differential diagnoses. Top-1 (equal to diagnostic sensitivity reported in the main results section), top-2 and top-3 sensitivities for diagnosing HCC were 73.6%, 90.8% and 96.6%, respectively in the internal test set and 81.5%, 89.0% and 93.6%, respectively in the external validation set (**S7 Table in S1 File**). This provides evidence that the AI system can characterize HCC with low miss rate.

The unique approach of our study is the development and testing of an AI system that can both detect and diagnose FLLs from USG still images. This novel AI system could automatically detect and classify FLLs without the need for human help for guiding the location of FLLs. Images with all ranges of qualities were included that help strengthen our findings on the practicality of using this AI method. We found that the CNN was able to handle such variation reasonably well. We believe that with more data, the performance of the AI system could be further improved.

The AI development flow can be divided into the following stages: 1) pre-clinical stage using single-site retrospective dataset, 2) validation on external cohorts, and 3) evaluating

usefulness of AI systems in real clinical settings by prospective or randomized-controlled trial study designs [29]. In this study, we validated the performance on external validation cohorts (i.e. 2<sup>nd</sup> stage of AI development flow) with satisfactory results. Currently, our AI system works off-line on still USG images. Since the ultimate goal is to implement an AI system in clinical practice, we are now incorporating USG videos as training materials to leverage our AI system to perform real-time analysis while a USG procedure is being performed.

## Conclusion

Given the structured training framework, the CNN has shown good performance for the detection and diagnosis of FLLs in USG images. HCCs can be detected and diagnosed with satisfactory performance. To fulfill our goal of assisting in the detection and diagnosis of FLLs during USG performed by non-radiologists, an AI system for real-time detection and analysis is warranted.

## Supporting information

**S1 File.**
(PDF)

## Acknowledgments

Authors thank the research team of the Department of Medicine, Faculty of Medicine, Chulalongkorn University for language-editing the final manuscript. We also thank Chrowarat Sritananun, Preeyanan Sae-lim and Chayanit Sukjaroen for their assistance in data acquisition.

## Author Contributions

**Conceptualization:** Thodsawit Tiyarattanachai, Sanparith Marukatat, Roongruedee Chaiteerakij.

**Data curation:** Thodsawit Tiyarattanachai, Terapap Apiparakoon, Sasima Sukcharoen.

**Formal analysis:** Thodsawit Tiyarattanachai, Terapap Apiparakoon, Sanparith Marukatat, Roongruedee Chaiteerakij.

**Funding acquisition:** Thodsawit Tiyarattanachai, Rungsun Rerknimitr, Roongruedee Chaiteerakij.

**Investigation:** Thodsawit Tiyarattanachai, Terapap Apiparakoon, Sanparith Marukatat, Sasima Sukcharoen, Nopavut Geratikornsupuk, Nopporn Anukulkarnkusol, Natthaporn Tanpowpong, Pamornmas Sarakul, Rungsun Rerknimitr, Roongruedee Chaiteerakij.

**Methodology:** Thodsawit Tiyarattanachai, Terapap Apiparakoon, Sanparith Marukatat, Roongruedee Chaiteerakij.

**Software:** Thodsawit Tiyarattanachai, Terapap Apiparakoon, Sanparith Marukatat.

**Supervision:** Sanparith Marukatat, Rungsun Rerknimitr, Roongruedee Chaiteerakij.

**Visualization:** Thodsawit Tiyarattanachai, Terapap Apiparakoon.

**Writing – original draft:** Thodsawit Tiyarattanachai.

**Writing – review & editing:** Thodsawit Tiyarattanachai, Terapap Apiparakoon, Sanparith Marukatat, Parit Mekaroonkamol, Rungsun Rerknimitr, Roongruedee Chaiteerakij.

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
