## [Decision Letter · Decision Letter 0]

29 Mar 2021

PONE-D-20-36551

Development and validation of artificial intelligence to detect and diagnose liver lesions from ultrasound images

PLOS ONE

Dear Dr. Chaiteerakij,

Thank you for submitting your manuscript to PLOS ONE. After careful consideration, we feel that it has merit but does not fully meet PLOS ONE’s publication criteria as it currently stands. Therefore, we invite you to submit a revised version of the manuscript that addresses the points raised during the review process.

We look forward to receiving your revised manuscript.

Kind regards,

Khanh N.Q. Le

Academic Editor

PLOS ONE

Journal Requirements:

Reviewers' comments:

Reviewer's Responses to Questions

**Comments to the Author**

1. Is the manuscript technically sound, and do the data support the conclusions?

Reviewer #1: Partly

Reviewer #2: Yes

2. Has the statistical analysis been performed appropriately and rigorously? 

Reviewer #1: Yes

Reviewer #2: Yes

3. Have the authors made all data underlying the findings in their manuscript fully available?

Reviewer #1: Yes

Reviewer #2: No

4. Is the manuscript presented in an intelligible fashion and written in standard English?

Reviewer #1: Yes

Reviewer #2: Yes

5. Review Comments to the Author

Reviewer #1: In this manuscript, the authors developed a convolutional neural network model for detection and classification of focal liver lesions in diagnostic ultrasound images. The current study had a larger training dataset and external testing data compared to existing literature on this topic. There are a lot of important information and data in the supporting document. Overall, the study is interesting and thorough; however, there are a number of issues that should be addressed.

1. Ultrasound image preprocessing, CNN model architecture and training process should be described in the Methods section.

2. Were all ultrasound images acquired with a curvilinear transducer? Were they all still images or from a video clip? Were the images resized to certain matrix size?

3. Besides markers on the images, were there any texts such as dimension measurement and annotations?

4. If some failure was due to dark images, could image intensity scaling be added into the image preprocessing step?

5. Consistent and standardized naming would be recommended. There was internal test which was sometimes called internal validation. It could be confusing with the validation in training process.

6. Could the training and internal test datasets from the same patients and same lesion? They should be different.

7. It’s not clear if the ground truth diagnosis labelling was based on MR or CT, or only pathology reports.

8. IoU cutoff of 0.2 appears to be loose. Was there any more justification on this cutoff value? Could IoU be optimized in the model?

9. How was the diagnostic accuracy calculated? Was it area under the ROC curve or something else? Without clarification, the first paragraph in the Discussion could be misleading.

Reviewer #2: In this work, the authors proposed a CNN for focal liver lesion detection from ultrasonography still images. Internal and external validation datasets were used to validate the proposed method. The whole study is complete. However, a lot of details were missing, which need further modifications.

Major comments:

1. The authors mentioned ‘detection’ and ‘diagnosis’ performance in the paper. However, it is hard to understand what ‘diagnosis’ metrics mean. ‘The AI system was… by one-versus-all method’. This paragraph is very difficult to understand. The authors need to rewrite the diagnosis task part.

2. The training phase section is not detailed enough. What’s the training epoch number for the model? Did the authors use validation datasets to choose the training epoch?

Minor comments:

1. Grammar errors exist throughout the paper, which need the authors to further address, e.g. ‘unweighted average method’.

2. The reference number should be put ahead of the period sign, not after.

3. Table 3. It is better to present it as a heat map, instead of a single table, to better visualize the results.

6. PLOS authors have the option to publish the peer review history of their article (what does this mean?). If published, this will include your full peer review and any attached files.

Reviewer #1: No

Reviewer #2: No

---

## [Author Response · Author response to Decision Letter 0]

16 Apr 2021

Reviewer #1: 

1. Ultrasound image preprocessing, CNN model architecture and training process should be described in the Methods section.

Response: We have added the CNN model architecture, ultrasound image preprocessing and training process in the Methods section on page 11-13 as follows:

AI system architecture

The AI framework used in this study was a convolutional neural network (CNN). CNNs are currently the preferred technique for several types of image analyses due to its structured layering characteristic that can detect complex features of the input images, where the shallow layers detect simple features such as dots and lines and the deeper layers detect more complex features, such as curves and loops. In the present study, we adopted a CNN architecture called RetinaNet which takes an image as input and creates a set of bounding boxes surrounding the FLL along with its class (predicted diagnosis) and its confidence in predicting that particular diagnosis. Confidence value range from 0 to 1, with a value of 1 being the most confident. The confidence threshold can be adjusted according to clinical relevance; for example, the confidence threshold may be lowered to increase the detection rate for HCC if needed in a certain patient population. The overall performance of the CNN, therefore, varies by different confidence thresholds. In this study, the confidence threshold was selected such that the F2 score was optimized on a tuning set, which was a subset of the training set (Details in S3 Appendix and S8 Fig). The selected confidence threshold was then used in both the internal test set and external validation cohorts. Since each diagnosis was independent, it was possible for RetinaNet to output multiple diagnoses for a single lesion. This approach resembled the usual practice of reporting differential diagnoses of FLLs by radiologists.

Ultrasound image preprocessing

During image preprocessing, all patient identification information and the peripheral areas in the USG images were cropped out. We identified the coordinates of fan-shaped USG region by ‘Sequence of Ultrasound Regions’ DICOM header, in order to ensure that the cropped image contained only the fan-shaped USG region where annotations and dimension measurements had been cropped out. We also removed markers which were made by sonographers in some images (S1 Appendix). The images were then resized to 1333 pixels wide and 800 pixels tall before inputted into the CNN.

AI system development process

Training phase

A supervised training method was implemented to train the AI system. In order to generate an image training dataset, pathology and/or MRI/CT reports were reviewed by experienced sonographers to identify labels, which were the locations and definitive diagnoses of FLLs in each USG image. A hepatologist (R.C.) subsequently verified the labels to ensure their accuracy. Images in the training set were fed into the AI system to train it to predict the location and diagnosis of the FLLs (Fig 2). 

RetinaNet codes were adopted from an open-source repository. The codes were then modified and optimized for analyzing USG images. In this work, RetinaNet was composed of backbone ResNet50 and the detection and diagnosis heads. The backbone ResNet 50 extracted the hierarchy of features, and the detection and diagnosis heads subsequently processed these features and outputted locations and diagnoses of FLLs.

The training was done in two main steps. First, the backbone ResNet50 was trained on a publicly-available image dataset called Microsoft Common Objects in Context (MS-COCO), which comprises 330,000 images of 1.5 million object instances. Subsequently, the whole CNN, both backbone and heads, was fine-tuned on our USG images in the training set. The CNN was trained for 500,000 iterations (25 epochs × 20,000 steps per epoch) on USG images. The initial learning rate was 0.0001. To enable the CNN to recognize diverse configurations of images and to maximize the number of training images, image augmentation was performed by horizontal translation, vertical translation, rotation, scaling, horizontal flipping, motion blur, contrast, brightness, hue and saturation adjustment at each iteration. The training hyperparameters are shown in the S8 Table. During training, a tuning set was used to monitor performance of the CNN. We selected an epoch that optimized mean average precision on the tuning set for final evaluation on the internal test set and the external validation set.

2. Were all ultrasound images acquired with a curvilinear transducer? Were they all still images or from a video clip? Were the images resized to certain matrix size?

Response: All ultrasound images were acquired with a curvilinear transducer. They were all still images. The images were resized to 1333 pixels wide and 800 pixels tall before inputted into the CNN.

We have added sentences in the Methods section as follows:

 In the Dataset subsection (page 9): “All data were still images taken as snapshots during ultrasound. They had been stored in Digital Imaging and Communications in Medicine (DICOM) format. All images were acquired using curvilinear transducers and allocated into 3 datasets: training set, internal test set and external validation set.”

 In the Ultrasound image preprocessing subsection (page 12): “The images were then resized to 1333 pixels wide and 800 pixels tall before inputted into the CNN.”

3. Besides markers on the images, were there any texts such as dimension measurement and annotations?

Response: There were dimension measurement and annotation texts in the periphery areas of the ultrasound images. These texts were cropped out during the image preprocessing step. 

We have added this explanation in the Methods section under Ultrasound image preprocessing subsection on page 12 as follows:

“We identified the coordinates of fan-shaped USG region by ‘Sequence of Ultrasound Regions’ DICOM header, in order to ensure that the cropped image contained only the fan-shaped USG region where annotations and dimension measurements had been cropped out.”

4. If some failure was due to dark images, could image intensity scaling be added into the image preprocessing step?

Response: We thank the reviewer for this insightful question. We have acknowledged this issue during our experiments. We tried using histogram equalization technique to scale image intensity. Unfortunately, the image quality was minimally improved by this technique. We found that brightness and contrast augmentation during the training step was more helpful in enabling the CNN to handle images with variation in light/dark conditions. Hyperparameters for brightness and contrast augmentation are shown in Table S8.

5. Consistent and standardized naming would be recommended. There was internal test which was sometimes called internal validation. It could be confusing with the validation in training process.

Response: We apologize for the inconsistency. We have changed the word “internal validation set” to “internal test set” throughout the manuscript.

6. Could the training and internal test datasets from the same patients and same lesion? They should be different.

Response: We apologize for the unclear explanation. We divided the dataset into the training and the internal test set based on patient level, i.e., images from the same patient were allocated to the same set. 

We have add sentences in the Method section, under the Dataset subsection on page 9 as follows:

“Allocation design ensured that all images from the same patient were assigned to the same set making the image sets independent of each other without any duplicated patients.”

7. It’s not clear if the ground truth diagnosis labelling was based on MR or CT, or only pathology reports.

Response: We again apologize for the unclear writing in the original manuscript. The pathology reports were considered first for ground truth diagnosis. If the pathology report was not available, MRI and CT reports were then considered. 

We have added this explanation to the Method section, under Dataset subsection on page 10 as follows:

“The definitive diagnoses of FLLs were verified using pathology and/or MRI/CT reports. Pathology reports were reviewed first. If not available, MRI and CT reports were then considered.”

8. IoU cutoff of 0.2 appears to be loose. Was there any more justification on this cutoff value? Could IoU be optimized in the model?

Response: In this study, we opted to use the IoU cutoff of 0.2 because FLLs in USG images often have indistinct boundary, especially for FFSs and FFIs. Nonetheless, our model achieved overall high IoU. In the internal test set, the median IoU was 0.788 (range: 0.202 – 0.978). In the external validation set, the median IoU was 0.781 (range: 0.201 – 0.970). Distribution of IoU values is summarized in Fig S3 and examples of images with different IoU values are illustrated in Fig S2.

We have added sentences to explain the reason for choosing the IoU cutoff of 0.2 in the Method section under the Performance evaluation metrics subsection on page 14 as follows:

 “ In this study, an IoU of greater than 0.2 was a cut-off for a correct detection by the CNN (S2 Appendix and S1 Fig). We opted to use this cutoff because FLLs in USG images often have indistinct boundaries, especially for FFSs and FFIs (S2 Fig).”

9. How was the diagnostic accuracy calculated? Was it area under the ROC curve or something else? Without clarification, the first paragraph in the Discussion could be misleading.

Response: We apologize for the unclear explanation. We calculated the accuracy by the following formula:

(TP+TN)/(TP+TN+FP+FN)

where TP, TN, FP and FN are the number of true positive, true negative, false positive and false negative classifications, respectively.

We have provided the formula used for calculating sensitivity, specificity, accuracy, positive predictive value and negative predictive value in the Methods section under Performance evaluation metrics subsection on page 14.

Additionally, to avoid ambiguity to the readers and to be consistent with the main results shown in Table 2, we have revised the first paragraph of the Discussion section on page 23 as follows:

 “ The CNN developed in our study using an advanced structured AI learning system demonstrated a consistently high diagnostic performance on USG images from both an internal test set and an external validation set. It achieved overall diagnostic sensitivity and specificity of 83.9% and 97.1% on the internal test set and 84.9% and 97.1% on the external validation set.”

Reviewer #2:

Major comments:

1. The authors mentioned ‘detection’ and ‘diagnosis’ performance in the paper. However, it is hard to understand what ‘diagnosis’ metrics mean. ‘The AI system was… by one-versus-all method’. This paragraph is very difficult to understand. The authors need to rewrite the diagnosis task part.

Response: We apologize for the unclear writing. We have rewritten the Diagnosis task subsection to describe the diagnosis metrics and explain the “one-versus-all” method on page xx as follows:

We used the following metrics to evaluate AI diagnostic performance:

sensitivity=TP/(TP+FN)

specificity=TN/(TN+FP)

accuracy=(TP+TN)/(TP+TN+FP+FN)

positive predictive value=TP/(TP+FP)

negative predictive value=TN/(TN+FN)

where TP, TN, FP and FN are the number of true positive, true negative, false positive and false negative diagnoses, respectively.

We used a “one-versus-all” method to evaluate diagnostic performance for each FLL diagnosis. For example, when evaluating diagnostic performance for HCC, other diagnoses were counted as a single non-HCC class:

〖sensitivity〗_HCC=〖TP〗_HCC/(〖TP〗_HCC+〖FN〗_HCC )

where 〖sensitivity〗_HCC is the diagnostic sensitivity for HCC.

〖TP〗_HCC is the number of true positive diagnoses for HCC, where the definitive diagnosis is HCC and the AI system correctly diagnosed the lesion as HCC.

〖FN〗_HCC is the number of false negative diagnoses for HCC, where the definitive diagnosis is HCC, but the AI system falsely diagnosed the lesion as either cyst, hemangioma, FFS or FFI.

2. The training phase section is not detailed enough. What’s the training epoch number for the model? Did the authors use validation datasets to choose the training epoch?

Response: We apologize for not providing the details of the training phase in the main manuscript. The training epoch number of the model was 25 (20,000 steps per epoch). We used a validation set to choose the training epoch. In our manuscript, the validation set was called ‘tuning set’ to avoid confusion with the external validation set.

We have added details regarding the training of the AI system in the Method section, under the Training phase subsection on page 12-13 as follows:

RetinaNet codes were adopted from an open-source repository. The codes were then modified and optimized for analyzing USG images. In this work, RetinaNet was composed of backbone ResNet50 and the detection and diagnosis heads. The backbone ResNet 50 extracted the hierarchy of features, and the detection and diagnosis heads subsequently processed these features and outputted locations and diagnoses of FLLs.

The training was done in two main steps. First, the backbone ResNet50 was trained on a publicly-available image dataset called Microsoft Common Objects in Context (MS-COCO), which comprises 330,000 images of 1.5 million object instances. Subsequently, the whole CNN, both backbone and heads, was fine-tuned on our USG images in the training set. The CNN was trained for 500,000 iterations (25 epochs × 20,000 steps per epoch) on USG images. The initial learning rate was 0.0001. To enable the CNN to recognize diverse configurations of images and to maximize the number of training images, image augmentation was performed by horizontal translation, vertical translation, rotation, scaling, horizontal flipping, motion blur, contrast, brightness, hue and saturation adjustment at each iteration. The training hyperparameters are shown in the S8 Table. During training, a tuning set was used to monitor performance of the CNN. We selected an epoch that optimized mean average precision on the tuning set for final evaluation on the internal test set and the external validation set.

Minor comments:

1. Grammar errors exist throughout the paper, which need the authors to further address, e.g. ‘unweighted average method’.

Response: We apologize for the grammar errors. We have sent the revised manuscript to our institution’s English editing service for correcting grammar errors. 

Additionally, we have added a subsection called ‘Calculation of overall detection rate and overall diagnostic performance’ in the Method section under Performance evaluation metrics subsection on page 15. This added subsection describes how we calculated overall performance, as follows:

Calculation of overall detection rate and overall diagnostic performance

After calculating detection and diagnostic performance metrics for each definitive diagnosis of FLLs, we pooled the performance results from the 5 FLL diagnoses into an overall performance result. Because the numbers of each FLL diagnosis in our dataset were imbalanced, overall performance, including overall detection rate, overall diagnostic sensitivity and specificity, were pooled by an unweighted average, to minimize the effect of imbalanced number of FLL diagnoses. For example,

〖DR〗_overall=(〖DR〗_HCC+〖DR〗_cyst+〖DR〗_hemangioma+〖DR〗_FFS+〖DR〗_FFI)/5

where 〖DR〗_overall is the overall detection rate.

〖DR〗_HCC, 〖DR〗_cyst, 〖DR〗_hemangioma, 〖DR〗_FFS and 〖DR〗_FFI are the detection rates for HCC, cyst, hemangioma, FFS and FFI, respectively.

〖sens〗_overall=(〖sens〗_HCC+〖sens〗_cyst+〖sens〗_hemangioma+〖sens〗_FFS+〖sens〗_FFI)/5

where 〖sens〗_overall is the overall diagnostic sensitivity.

〖sens〗_HCC, 〖sens〗_cyst, 〖sens〗_hemangioma, 〖sens〗_FFS and 〖sens〗_FFI are the diagnostic sensitivities for HCC, cyst, hemangioma, FFS and FFI, respectively.

2. The reference number should be put ahead of the period sign, not after.

Response: We thank the reviewer for this comment. We have moved the reference number ahead of the period sign, as suggested by the reviewer. 

3. Table 3. It is better to present it as a heat map, instead of a single table, to better visualize the results.

Response: We thank you the reviewer for this suggestion. Table 3 has been revised and presented as a heatmap accordingly.

---

## [Decision Letter · Decision Letter 1]

25 May 2021

Development and validation of artificial intelligence to detect and diagnose liver lesions from ultrasound images

PONE-D-20-36551R1

Dear Dr. Chaiteerakij,

We’re pleased to inform you that your manuscript has been judged scientifically suitable for publication and will be formally accepted for publication once it meets all outstanding technical requirements.

Kind regards,

Khanh N.Q. Le

Academic Editor

PLOS ONE

Additional Editor Comments (optional):

Reviewers' comments:

Reviewer's Responses to Questions

**Comments to the Author**

1. If the authors have adequately addressed your comments raised in a previous round of review and you feel that this manuscript is now acceptable for publication, you may indicate that here to bypass the “Comments to the Author” section, enter your conflict of interest statement in the “Confidential to Editor” section, and submit your "Accept" recommendation.

Reviewer #1: All comments have been addressed

Reviewer #2: All comments have been addressed

2. Is the manuscript technically sound, and do the data support the conclusions?

Reviewer #1: Yes

Reviewer #2: Yes

3. Has the statistical analysis been performed appropriately and rigorously? 

Reviewer #1: Yes

Reviewer #2: Yes

4. Have the authors made all data underlying the findings in their manuscript fully available?

Reviewer #1: Yes

Reviewer #2: No

5. Is the manuscript presented in an intelligible fashion and written in standard English?

Reviewer #1: Yes

Reviewer #2: Yes

6. Review Comments to the Author

Reviewer #1: (No Response)

Reviewer #2: The authors have addressed my comments. it is an interesting work in the area of ultrasound imaging.

7. PLOS authors have the option to publish the peer review history of their article (what does this mean?). If published, this will include your full peer review and any attached files.

Reviewer #1: No

Reviewer #2: No

---

## [Editor Report · Acceptance letter]

31 May 2021

PONE-D-20-36551R1 

Development and validation of artificial intelligence to detect and diagnose liver lesions from ultrasound images 

Dear Dr. Chaiteerakij:

I'm pleased to inform you that your manuscript has been deemed suitable for publication in PLOS ONE. Congratulations! Your manuscript is now with our production department. 

Kind regards, 

on behalf of

Dr. Khanh N.Q. Le 

Academic Editor

PLOS ONE